# Identification of Antioxidant Peptides in Enzymatic Hydrolysates of Carp (*Cyprinus Carpio*) Skin Gelatin

**DOI:** 10.3390/molecules24010097

**Published:** 2018-12-28

**Authors:** Joanna Tkaczewska, Michal Bukowski, Paweł Mak

**Affiliations:** 1Department of Animal Product Technology, Faculty of Food Technology, University of Agriculture in Cracow, Balicka 122 Street, 30-149 Cracow, Poland; 2Department of Analytical Biochemistry, Faculty of Biochemistry, Biophysics and Biotechnology, Jagiellonian University, Gronostajowa 7 Street, 30-387 Krakósw, Poland; m.bukowski@uj.edu.pl (M.B.); pawel.mak@uj.edu.pl (P.M.); 3Małopolska Centre for Biotechnology, Jagiellonian University, Gronostajowa 7A Street, 30-387 Kraków, Poland

**Keywords:** gelatin, bioactive peptides, fish skin, antioxidant, Ala-Tyr, identification

## Abstract

The protein by-products from carp (*Cyprinus carpio*) are normally discarded as industrial waste during fish processing. The objective of this study was to identify and characterise the peptides with a potential antioxidant activity that are released from carp skin proteins during hydrolysis by the Protamex enzyme mixture. This study shows that a hydrolysate of carp skin gelatin and its reversed-phase chromatography fractions have strong in vitro antioxidant properties. Among these fractions, the alanine-tyrosine (Ala-Tyr) dipeptide was identified as the major compound with high antioxidant potential. The peptide has good stability during in vitro enzymatic digestion assay and can inhibit the angiotensin-converting enzyme (ACE). In conclusion, our study proves that both the unfractionated hydrolysate of carp skin gelatin and the above-mentioned Ala-Tyr dipeptide represents attractive novel compounds for the formulation of antioxidant foods.

## 1. Introduction

In the human body, the imbalance between oxidation processes involving free-radical production and the efficiency of antioxidative mechanisms leads to the development of many lifestyle diseases. Substances that inhibit activity of free radicals retard cell ageing and may play a significant role in the prevention of degenerative diseases occurring at an older age [1]. Antioxidants are also used as preservatives for food. Oxidative transformation in food products is a crucial problem in food technology because it causes deterioration of the sensory quality of products and reduces their nutritional value. On the contrary, extensive literature indicates that the use of synthetic antioxidants in food could be the cause of many disorders amongst consumers, such as liver damage or tumours [2]. This fact has led to the identification of new sources of antioxidants in food ingredients already present in the market.

Production by capture fisheries in 2017 remained stable at 90.4 million tonnes. Nonetheless, because a significant proportion of wild catches is utilised for fish feed, aquaculture’s share in direct human consumption is now 55% and increases with each passing year. Domestically produced and cheap species, such as carp (*Cyprinus carpio*), are still an important staple in emerging markets across East and Southeast Asia [3]. Carp (*Cyprinus carpio L*), as a freshwater fish, is one of the most-willingly bred species in the world. It is characterized by a fast growth rate and good fodder usage [4]. In 2015, the production of carp in Europe totalled 4,328,083 tonnes and is starting to decline [5]. The main reason for this decline in the market demand for carp is that its traditional offer is mostly in the form of whole fish. To increase the range of products from carp, manufacturers are looking for alternatives to fresh carp, which is why these manufacturers are dynamically developing [4]. Processing plants produce carp mainly in the form of fillets without skin. The commercial use of carp skins, which until now have been treated as fish industry waste, can bring tangible economic benefits. Carp skin is a good source of collagen (containing 16%) [6]. This collagen can be processed into protein hydrolysates with antioxidant as well as other functional properties. 

As reported in the literature, protein hydrolysates of gelatin from fish skin often have antioxidant properties and can be an alternative to synthetic antioxidants [7]. Antioxidant peptides naturally present in food have a potential for use as factors that prevent adverse texture development, improve organoleptic properties, and cause functional or nutritional changes in food products owing to oxidation as well as for the production of pharmaceuticals and cosmetics with lipids. The application of such peptides may not only extend the lifespan of products (e.g., meat products), but also increase both their attractiveness for consumers and their availability [8]. According to our preliminary studies, antioxidative properties are also present in a hydrolysate prepared from the carp skin gelatin. The objective of this study was to identify and characterise the peptides with a potential antioxidant activity that are released from carp skin proteins during hydrolysis by the Protamex enzyme mixture.

## 2. Results and Discussion

The composition and amino acid profile of the carp skin gelatin hydrolysate. The composition of freeze-dried hydrolysate of carp skin gelatin is given in Table 1. 

The hydrolysate had a high protein concentration and low fat and water contents. According to Chalamaiah et al. (2012) [9], the protein content of fish protein hydrolysates ranges between 60% and 90% of the total composition; this finding is consistent with our results. The high protein content of carp skin hydrolysates highlights its potential use in the functional food industry.

The low-fat content of the protein hydrolysates of carp gelatin is a big advantage because it correlates with their low susceptibility to oxidative processes. This is an important factor for the storage and use of protein hydrolysates in the industry. Fat is a medium for fish taste and aroma and leads to colour changes: The low concentration of fat in protein hydrolysates of the carp gelatin increases their quality and practical potential as well as positively affects the sensory characteristics of the obtained hydrolysates. It can be assumed that the hydrolysates with an insignificant fish smell and taste will be attractive and intentionally used as bioactive compounds for functional foods. The high ash content could be caused by the addition of an acid and alkali required for pH adjustment and control during hydrolysis.

The amino acid profile of the carp gelatin hydrolysates is presented in Table 1. The predominant amino acids were Gly, Pro and Ala, which constituted approximately 28.12%, 13.60% and 11.33% of the total amino acid content, respectively. These data are in an agreement with our previous findings that Gly, Pro and Ala are some of the most abundant amino acid residues in gelatin from carp skin [6]. The high proportion of Gly, Pro and Ala has also been found in many ACE-inhibitory peptides [10].

It is worth noting that the hydrophobic amino acids constituted 60.67% of all the amino acids comprising the hydrolysate. According to Centenaro et al. (2014) [11], for hydrolysates of proteins and peptides, the greater the hydrophobicity, the higher the lipid solubility, and, therefore, the higher the antioxidant activity. Thus, the result on the amino acid profile served as a preliminary factor to evaluate the potency of bioactivity of carp skin–derived peptides. The rich source of potentially functional amino acids identified in the protein hydrolysates was therefore assumed to contribute to the antioxidant and ACE-inhibitory activities.

### 2.1. Isolation and Purification of the Antioxidant Peptides

The antioxidant activity of proteins and peptides is not contingent on only a single mechanism because proteins contain various amino acids with different biological properties. Some antioxidant components are more effective as radical scavengers or lipid peroxidation inhibitors, whereas some others are metal-chelating or reducing agents [7]. Hence, this study involved three in vitro methods, such as 2,2-diphenyl-1-picrylhydrazyl (DPPH)radical scavenging, ferric reducing antioxidant power (FRAP), and metal-chelating activity assays.

The gelatin hydrolysate from carp skin obtained as a result of digestion with the Protamex enzyme mixture is characterised by good antioxidant properties. Fu and Zhao (2015) [12] reported that the hydrolysis of gelatin gave rise to a noticeable increase in antioxidant capacity. In addition, according to these authors, the enzymatic hydrolysis of gelatin by various proteases can generate certain peptide sequences responsible for different bioactivities, including antioxidant activity. 

The antioxidant peptide from the carp skin hydrolysate was purified via a simple two-stage procedure comprising gel filtration and reversed-phase high performance liquid chromatography (RP-HPLC).

Measurements of antioxidant activity by the FRAP method for fractions from gel filtration indicated that this activity was distributed to various degrees among all the collected fractions (Figure 1). 

The total antioxidant activity of the gelatin hydrolysate measured by the FRAP method is a function of the activity of its individual fractions. Nonetheless, during gel filtration, the most active fractions were eluted at the end of the chromatogram (fractions No. 60 and 61, Figure 1). According to the absorbance at 220 nm, this fraction was marked as F8 and was equivalent to ca. 4% of the initial separated material (and responsible for ca. 10–13% of the total antioxidant activity of the hydrolysate). This fraction was next separated by reversed-phase HPLC into three major sub-fractions (F8-1, F8-2, and F8-3; Figure 2A).

It can be observed that the peak with a retention time of 9.5–11.0 min (marked as F8-2) had a particularly high antioxidant activity as measured by the FRAP method (26.64 μM Trolox/mg protein, Figure 2B). The remaining fractions had more than 10-fold less antioxidant activity according to this method. The FRAP method is based on a single electron transfer (SET) reaction between an oxidant and a free radical [13]. Consequently, the results of the FRAP method regarding RP-HPLC fractions suggested that fraction F8-2 is a strong natural antioxidant, acting according to the single electron transfer (SET) mechanism. 

On the contrary, all the fractions collected during gel filtration featured a low capacity for iron ion chelation (<10%; Table 2). 

This result most likely indicates that the chelation of iron ions by the output hydrolysate at 64.01% is a process for which different peptides are responsible cumulatively, and that it is completely independent from the antioxidant activity measured by the FRAP method.

The observed stronger chelating activity of hydrolysed gelatin from carp skin, as compared to the activity of fractions obtained by gel filtration, may indicate a synergistic action of the peptides present in the hydrolysate. These results are consistent with other reports, which have shown that unfractionated protein hydrolysates are better metal ion chelators than their fractionated counterparts [14]. The measurement of the chelating capacity toward iron ions in the F8-2 subfraction obtained after RP-HPLC revealed that despite the high activity measured by the FRAP method, this fraction did not have the ability to chelate iron ions.

The DPPH assay of the scavenging of free radicals by the fractions collected during the gel filtration indicated that such activity observed in the initial hydrolysate (23.76%) was rather a cumulative result of many different fractions. Fractions F1, F3, F4, and F7 (Figure 1) did not manifest the ability to scavenge free radicals (0%), but the other tested fractions had the ability to capture free radicals at the level of less than 13% (Table 2). The measurement of scavenging of free radicals by the F8-2 subfraction obtained after RP-HPLC suggested that despite the high activity detected by the FRAP method, this fraction did not have the ability to scavenge free radicals.

Several methods have been developed to evaluate the total antioxidant activity of food and animal tissues. Among them, the FRAP method, DPPH method, and metal-chelating assay are the most representative approaches frequently used in various studies. Each of the described tests of antioxidant activity has a specific mechanism of action with its own benefits and drawbacks; however, in the absence of a universal method that can produce clear results on a given product, the best solution is to employ multiple methods at the same time. According to Zou et al. (2016) [15], due to the different underlying mechanisms between evaluation assays, the values or variation trends obtained by different assays differ widely, even for the same amino acid sequence and composition.

According to Chen et al. (2012) [2], purification of a protein hydrolysate can greatly increase its antioxidant activity. Nevertheless, the results obtained in our study do not confirm this rule. The antioxidant ability of the purified F8-2 fraction—as measured by the FRAP method in comparison with the unfractionated hydrolysate—increased over 14-fold. In contrast, after the measurement by the DPPH method and by the assay of chelation of iron ions, the antioxidant ability of the purified F8-2 fraction significantly decreased (two-fold and 10-fold, respectively). This means that the total antioxidant activity of the hydrolysate from carp skin corresponds to a cumulative effect dependent on different peptides, not only on one purified peptide. These assumptions were confirmed by Sarmadi et al. (2010) [16], who demonstrated that the overall antioxidative activity of a protein hydrolysate must be ascribed to the integrative effects of several compounds rather than to the individual actions of peptides. 

As a result of the presented research, it was found that the F8-2 fraction is a strong antioxidant, acting in accordance with the SET mechanism; therefore, it was further analysed in detail to identify the peptide responsible for this activity.

### 2.2. Biochemical Characterisation of the Purified Peptide

The active peak being eluted at 10 min (Fraction F8-2, Figure 2A) was subjected to N-terminal sequencing and yielded only a two-amino acid-long sequence: Ala-Tyr. The analysis of this peptide by mass spectrometry revealed a spectrum containing a dominant peak at *m*/*z* = 253.2, equivalent to the final mass of 252.2 Da (Figure 3A). This mass is in good agreement with the theoretical mass of the Ala-Tyr dipeptide, which amounts to 252.27 Da. The MS/MS spectrum of the 253.2 ion independently confirmed the sequence of the Ala-Tyr peptide (Figure 3B).

According to Kou et al. (2013) [17], the lower-molecular-weight peptides (molecular mass from 200 to 3000 Da) are the most efficient antioxidants, and bioactive peptides usually contain less than 20 amino acid residues per molecule. The peptides with lower molecular weight have a better chance to cross the intestinal barrier and to exert biological effects. The Ala-Tyr peptide with the greatest antioxidant activity isolated during our study had a low molecular weight (253.2 Da), much lower than that of the antioxidant peptides isolated until now from fish gelatins [18,19,20]. 

The antioxidative activities of peptides are higher than those of free amino acids; this phenomenon is attributed to the unique chemical and physical properties contributed by the amino acid sequences, especially the stability of the resultant radicals that do not initiate or propagate further oxidative reactions [17]. It is also estimated that the presence of amino acids, including Ala and Tyr, may contribute to the antioxidant activities of peptides [21]. These results underscore the importance of amino acid composition and peptide size for the antioxidative potential of peptides. In the present study, the identified peptide (Ala-Tyr) contained the amino acids that have an antioxidant activity. Thus, the carp skin gelatin hydrolysate has a great potential as a natural antioxidant ingredient for food and pharmaceutical applications.

### 2.3. Confirmation of the Biological Activity of the Synthetic Ala-Tyr Peptide

To evaluate the antioxidant activities of the Ala-Tyr dipeptide, the DPPH, FRAP, and metal-chelating assays of the synthetic peptide were conducted. The results are given in Table 3.

These results clearly prove that the synthetic peptide, AY, has good antioxidative activity. The FRAP value was the highest (89.23 μM Trolox/mg sample), and its DPPH-scavenging ability was the second highest (38.69%). On the contrary, the synthetic peptide showed no ability to chelate iron ions (3.09%). AY had a high antioxidant activity possibly due to the hydrophobic amino acid content. As reported in the literature, hydrophobic amino acids in peptides can obviously improve oxidation resistance [22]. Sae-Leaw et al. (2017) [23] noted that the presence of hydrophobic amino acids, such as alanine, in the gelatin peptides from seabass skins increases their antioxidant properties. According to these authors, the accessibility of hydrophobic antioxidant peptides to hydrophobic cellular targets, such as the polyunsaturated chain of fatty acids, of biological membranes could be augmented, thereby facilitating the interaction with radical species. According to Delgado et al. (2016) [24], peptides with Tyr residues have the highest antioxidant activity. Additionally, the presence of Tyr at the C-terminal position enhances the scavenging activity.

The results indicate that AY may contribute to the antioxidant potential. Nevertheless, other studies on AY as an antioxidant peptide have not yet been reported. 

It has been shown that hydrophobic amino acid residues, such as leucine, valine, alanine, tyrosine, phenylalanine, or tryptophan, can act as competitive angiotensin-1-converting enzyme (ACE) inhibitors as they preferably bind the catalytic sites of ACE [25]. Furthermore, according to data from the literature, after oral administration in spontaneously hypertensive rats, dipeptides with tyrosine at the C-terminal caused slow, but prolonged reduction of systolic blood pressure [26]. Thus, another test was carried out to determine *in vitro* inhibition of ACE by the identified peptide, as shown in Figure 4. 

The synthetic peptide at the concentration of 10 mg/mL had more than a two-fold better ability to inhibit ACE than did the hydrolysate of carp skin at a concentration of 50 mg/mL (27.17% and 66.44%, respectively). The peptide, AY, contains an aromatic amino acid, which may contribute to the increased potency by enhancing hydrophobic interactions with the enzyme molecule [27].

Dipeptide Ala-Tyr, which is derived from Cyprinus carpio skin gelatin hydrolysate, possesses antihypertensive activities; along with similar dipeptides, it has received considerable attention for its antihypertensive effects in vivo [28,29,30]. Also, in their research, Nakahara et al. [31] isolated nine dipeptides that had strong ACE-inhibitory properties from Soy Sauce-Like Seasoning, most of these peptides having an alanine in the N-terminal end, and/or tyrosine in the C-terminal end. The peptide, AY, with anti-hypertension activity has been isolated by Yang et al. in 2007 [32] from corn gluten meal. In their tests, with the participation of animals, it was found that the minimum effective oral dose of peptide AY to reduce blood pressure by 9.5 mm Hg was 50 mg/mL. As a result of our research, it can be stated that the hydrolysate of gelatin from carp skin is the source of a peptide with anti-hypertension properties.

### 2.4. The Effect of Gastrointestinal Proteases on the Antioxidant Activity of the Synthetic Peptide AY and Gelatin Hydrolysate

The obtained carp gelatin hydrolysate and synthetic peptide were next subjected to simulated gastrointestinal digestion with pepsin and pancreatin to evaluate their stability and antioxidant activity after digestion. Only weak changes were found after digestion (Table 3). The antioxidant activity of the hydrolysate and of the peptide tested with the DPPH method did not differ statistically significantly; however, the antioxidant activity of digested samples evaluated by the FRAP method was significantly lower. Similar results were obtained by Harnedy et al. (2017) [33], who digested antioxidant peptides isolated from the *Palmaria palmata* seaweed. 

The ability of the gelatin hydrolysate from carp skin to chelate iron ions was stable after in vitro digestion, whereas the synthetic peptide after digestion manifested a many-fold higher chelating capacity than before digestion (73.25% versus 3.09%, respectively). It can be assumed that as a result of the action of digestive enzymes, further hydrolysis of the peptide occurred, and free amino acids, Ala and Tyr, were formed. According to Gülçin (2007) [34] and Wu et al. (2003) [35], l-Tyr and Ala can chelate metal ions (at 60% and 32%, respectively); this observation may explain the increase in this capacity after peptide digestion.

These results indicate that the hydrolysate and synthetic peptide, Ala-Tyr, could effectively retain the overall antioxidant effect after in-vitro digestion. Future work should concentrate on the examination of the ability of this peptide to reduce oxidative stress in living tissues.

## 3. Materials and Methods

### 3.1. Extraction and Hydrolysis of Gelatin from Carp Skin

The carp skin was obtained from the Sona Sp. z o.o. fish processing plant (Koziegłowy, Poland), where it was treated as waste after fish filleting. The skin was cleaned up to remove adjacent tissue and was ground up (Mado MEW 613). Gelatin was extracted by the method of Duan et al. (2011) [36] with a modification introduced by Tkaczewska et al. (2018) [6]. The Protamex^®^ (Novozymes) enzyme preparation was used for gelatin hydrolysis. In brief, 10 g of the lyophilised protein was dissolved in 150 mL of water, heated to 50 °C, and then pH 7 was attained with 1 M HCl. The addition of the enzyme preparation represented 2% (*w*/*w*) of the protein content. Based on our preliminary research, the reactions were carried out for 3 h. For the first 15 min, pH was constantly monitored and adjusted with 1 M NaOH, then pH was corrected every 15 min. The reaction was completed by incubating hydrolysates at 95 °C for 15 min, then the samples were cooled down in an ice bath and centrifuged at 8000× *g* for 15 min at 10 °C.

### 3.2. The Composition and Amino Acid Profile of the Obtained Hydrolysate

Water, lipids, ash, and protein contents were determined by the Association of Official Analytical Chemists (AOAC)-recommended methods [37].

Before amino acid analysis, 30 mg of enzymatic hydrolysate was additionally hydrolysed using 4 mL of 6 M HCl and 15 µL of phenol at 110 °C for 24 h. The sample was sealed in a nitrogen atmosphere during the process of hydrolysis. The obtained hydrolysate was passed through a syringe filter with a pore diameter of 45 µm and dried under a constant stream of nitrogen. Next, 10 mL of the prepared solution was derivatised by mixing with 70 µL of borate buffer (pH 8.2–9.0) and 20 µL of 6-aminoquinolyl-*N*-hydroxysuccinimidylcarbamate (Waters, Milford, MA, USA) in an acetonitrile solution (3:1, *w*/*v*). The standards were derivatised in the same manner as the samples.

The amino acid analysis was carried out using a Dionex Ultimate 3000 HPLC system (Thermo Fisher Scientific, Waltham, MA, USA) equipped with an LPG-3400 SD four-channel gradient pump, WPS 3000 TSL autosampler, and FLD 3400RS four-channel fluorescent detector. Analysis was performed on a Nova-Pak C18, 4 µm (150 × 3.9 mm) column (Waters, Milford, MA, USA). The elution buffers were (A) acetate-phosphate buffer and (B) 60:40 acetonitrile/water. The separation temperature was set to 37 °C, and detection settings were as follows: Excitation at 250 nm and emission at 395 nm wavelengths. The quantitative analysis was performed by 1-point calibration with analytical standards (50 pmol for each concentration). The quantification of tryptophan was performed in accordance with the Buraczewska and Buraczewski (1984) [38]. 

### 3.3. Fractionation of an Enzymatic Hydrolysate and Purification of the Active Peptide

The dry powder of the hydrolysate was dissolved in ultrapure water at a concentration of 100 mg/mL, centrifuged at room temperature for 5 min at 20,000× *g*, and passed through a 0.45 μm filter. Gel filtration was performed in ultrapure water on a HiLoad 16/600 Superdex^®^ 30 pg column (GE Healthcare, Chicago, IL, USA) at a 0.3 mL/min flow rate. Then, 1 mL of the hydrolysate solution was loaded on the column at a specific time point. Collected fractions (1.5 mL) were subjected individually to measure absorbance at 220 nm and 280 nm on a NanoDrop 2000 spectrophotometer (Thermo Fisher Scientific, Waltham, MA, USA), and independently, to estimate the antioxidant activity. The most active fraction was lyophilised, dissolved in 0.1% (*v*/*v*) trifluoroacetic acid (TFA), and subjected to reverse-phase high pressure liquid chromatography (RP-HPLC) using an UltiMate 3000 chromatograph (Dionex™/Thermo™, Sunnyvale, CA, USA). The separation was performed on a Discovery C18 250 × 4.6 mm column (Sigma-Aldrich, Poznan, Poland) at a 1 mL/min flow rate by means of two buffers: (A) 0.1% TFA (*v*/*v*) and (B) 0.07% TFA in 80% acetonitrile (both *v*/*v*). The linear gradient from 0% to 60% of buffer B was set up in 15 min, while the spectrophotometric detection was carried out at 220 nm and 280 nm. The collected fractions were evaporated in a vacuum centrifuge and redissolved in ultrapure water.

### 3.4. Biochemical Characterisation of the Purified Peptide

The N-terminal amino acid sequence was determined on an automatic protein sequencer, PPSQ-31A (Shimadzu Corp., Kyoto Prefecture, Japan). Mass-spectrometric analyses were performed by means of an AmaZon ETD spectrometer (Bruker Corp. Billerca, MA, USA) equipped with an electrospray ion source and an on-line nano-LC chromatograph (Proxeon A/S, Odense, Denmark). 

### 3.5. Determination of the Antioxidant Activities

Ferric reducing ability of plasma (FRAP) method.

Determination of the reducing potential of samples was performed according to the method described by Khantaphant and Benjakul (2018) [39] with some modifications. The oxidant in the FRAP assay consisted of acetate buffer (pH 3.6), a ferric chloride solution (20 mM), and a 2,4,6-tripyridyl-s-triazine solution (10 mM TPTZ in 40 mM HCl) at 10:1:1 (*v*/*v*/*v*), respectively, and was freshly prepared on the day of analysis. Thus, 100 μL of an appropriately diluted sample was added to 900 μL of the above FRAP solution in microfuge tubes and vortexed. The tubes were incubated at 37 °C for 30 min in the dark, and after that, absorbance was measured at 593 nm. Greater absorbance represented higher reducing power of a sample. Some of the results were expressed as Trolox equivalents (in μmol) of activity per 1 mg of the hydrolysate.

#### 2,2-diphenyl-1-picrylhydrazyl (DPPH) method

The DPPH free radical quenching was determined by the method of Wu et al. (2003) [31] with a modification described by Borawska et al. (2016) [40]. Equal volumes of the sample and DPPH solution (0.15 mM) were mixed in 95% ethanol. The absorbance of the mixture was then measured at 517 nm. Lower absorbance represented a higher DPPH-scavenging activity. The scavenging effect was calculated using the formula: $$DPPH\;radical\;scavenging\;\left(\%\right)=\;\frac{A_{blank}-\;A_{sample}\;}{A_{blank}}\times 100\%$$

Ferrous-ion–chelating abilities.

The ability of the sample to chelate ferrous ions was assessed via the method of Guo et al. (2001) [41]: 0.5 mL of a sample was first mixed with 4.5 mL of distilled water. It was then mixed with 0.1 mL of 2 mM FeCl_2_ and then 0.2 mL of 5 mM ferrozine. After 10 min, absorbance of the reaction mixture was measured at 562 nm. The percentage of ferrous-ion–chelating ability was calculated as follows: $$Chelating\;ability\;\left(\%\right)=1-\;\frac{\;A_{sample}\;}{A_{blank}}\times 100\%$$

### 3.6. Measurement of ACE-Inhibitory Activity of the Synthetic Peptide and Hydrolysate 

A synthetic alanine-tyrosine (Ala-Tyr) dipeptide was purchased from Sigma-Aldrich (cat. No. A4003). The purity of the synthesized peptides was 100% as determined by thin layer chromatography. 

The ACE-inhibitory activity was measured by a spectrophotometric assay of Nasri et al. (2013) [42] with some modifications. In brief, 80 μL of a synthetic peptide (10 mg/mL) or hydrolysate (50 mg/mL) solution was added to 200 μL of 5 mM hippuryl-l-histidyl-l-leucine (HHL), and then preincubated for 3 min at 37 °C. The peptide and HHL were dissolved in 100 mM borate buffer (pH 8.3) containing 300 mM NaCl. The reactions were then initiated by adding 20 μL of 0.1 U/mL ACE from rabbit lungs (Sigma-Aldrich, Poznan, Poland), prepared in the same buffer. After incubation at 37 °C for 30 min, the enzymatic reactions were stopped by addition of 250 μL of 0.05 M HCl. The released hippuric acid (HA) was extracted with ethyl acetate (1.7 mL). The upper layer of the solution was evaporated at 95 °C under a constant flow of nitrogen. The residue was dissolved in 1 mL of distilled water, and the absorbance of the extract at 228 nm was determined on a spectrophotometer (UV/VIS, HeliosWaltham, MA, USA). The ACE-inhibitory activity was calculated using the equation: $$ACE\;inhibition\;\left(\%\right)=\;\frac{A_{1}-\;A_{2}\;}{A_{1}-\;A_{3}}\times 100\%$$ where A_1_ is the absorbance of the ACE solution without an inhibitor (carp gelatin protein hydrolysate), A_2_ denotes the absorbance of the tested sample of carp protein hydrolysate, and A_3_ indicates the absorbance of the HHL solution (a buffer was added instead of the ACE solution and sample).

### 3.7. Effects of Gastrointestinal Proteases on the Antioxidant Activity of the Synthetic Peptide and Gelatin Hydrolysate

An in vitro digestion model system involving the enzymes similar to those in the upper gastrointestinal digestive tract of humans was employed as described by Teixeira et al. (2016) [43]. pH of the synthetic peptide or hydrolysate solutions (30 mg/mL in water) was adjusted to 2.0 with HCl (1 M), and pepsin 40 mg per g of protein (Sigma-Aldrich, Poznan, Poland)] was added. After incubation at 37 °C for 1 h, pH of the solution was adjusted to 5.3 with 0.9 M NaHCO_3_. Pancreatin 40 mg per g of protein (Sigma-Aldrich, Poznan, Poland)] was added, pH was adjusted to 7.5 (1 M NaOH), and the mixture was incubated at 37 °C for 2 h. At the end of the digestion, both enzymes were inactivated by heating in boiling water for 10 min. The digested peptide and hydrolysate solution were centrifuged (11,000× *g*; 15 min), and the supernatant was lyophilised. The antioxidant activities of the digested peptide and hydrolysate were measured after digestion.

### 3.8. Statistical Analysis

All the analyses were performed in triplicate and the data were subjected to a statistical analysis in the STATISTICA 12.0 software (Dell Software, Tulsa, OK, USA). Significance of the differences between groups was established by one-way analyses of variance (ANOVA) and Tukey’s *post hoc* test (*p* < 0.05). The results are presented as average ± standard deviation.

## 4. Conclusion

The presented study shows that the carp skin gelatin hydrolysate and its RP-HPLC fractions have strong in vitro antioxidant properties, as evidenced by their high FRAP value. Nonetheless, according to metal chelation and DPPH-scavenging assays, fractionation of the hydrolysate attenuated its antioxidant properties. In the hydrolysate, the Ala-Tyr dipeptide was identified, and it is responsible for the high antioxidant activity tested using the FRAP method. The peptide has good stability during in vitro digestion assay. Given that a negative effect of fractionation was observed for natural free radicals, the unfractionated hydrolysate of carp skin gelatin may be a better alternative as an ingredient for formulation of antioxidant foods in comparison with the fractionated peptides. The small size (250 Da) of the dipeptide discovered in this work may contribute to its good intestinal absorption and potency as an in vivo antioxidant. Future work will concentrate on the examination of the ability of this peptide to reduce oxidative stress in living tissues. The results of this study can aid in the development of practical applications of fish gelatin peptides as antioxidants.

## Figures and Tables

**Figure 1 molecules-24-00097-f001:**
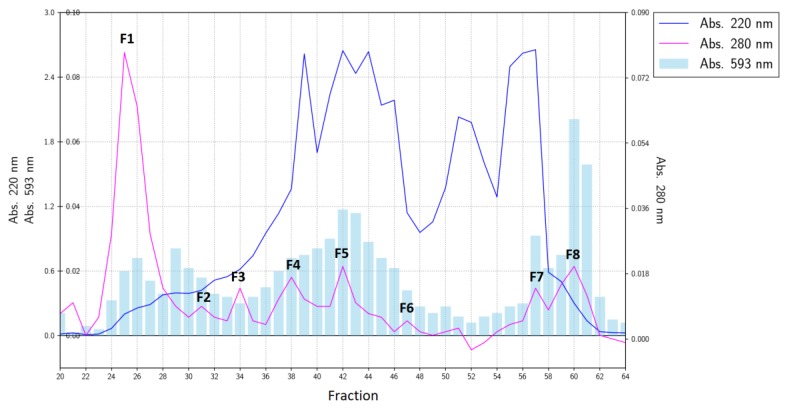
The elution profile of a carp skin gelatin hydrolysate separated by gel filtration and the ferric reducing antioxidant power (FRAP) of the fractions.

**Figure 2 molecules-24-00097-f002:**
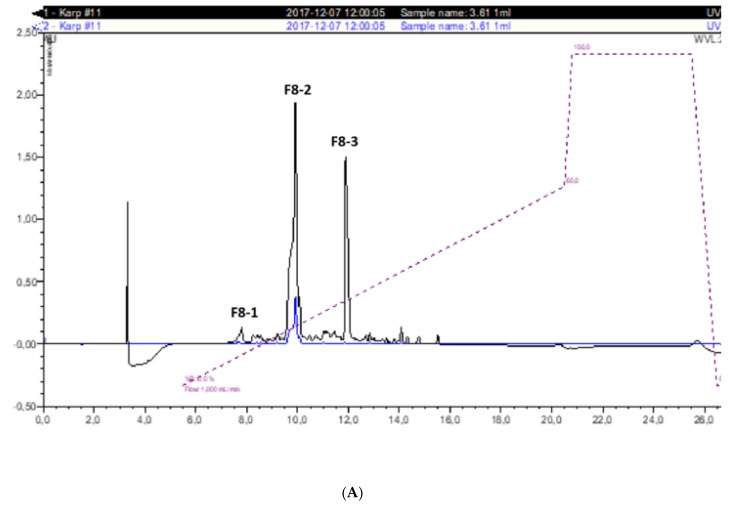
(**A**) The reversed-phase HPLC pattern of the F8 fraction obtained by gel filtration and (**B**) the FRAP activity of the eluted peaks expressed in μM Trolox/mg protein.

**Figure 3 molecules-24-00097-f003:**
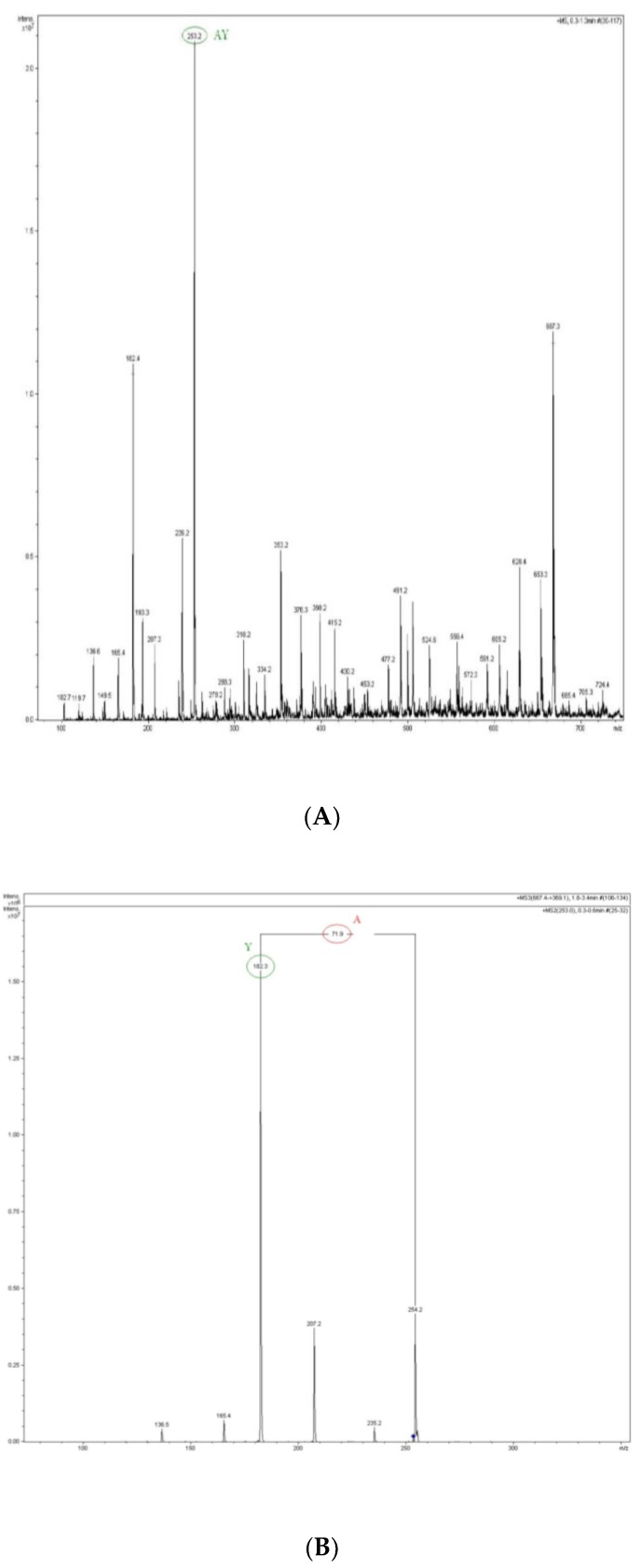
The MS (**A**) and MS/MS spectrum (**B**) of the peptide from fraction F8-2.

**Figure 4 molecules-24-00097-f004:**
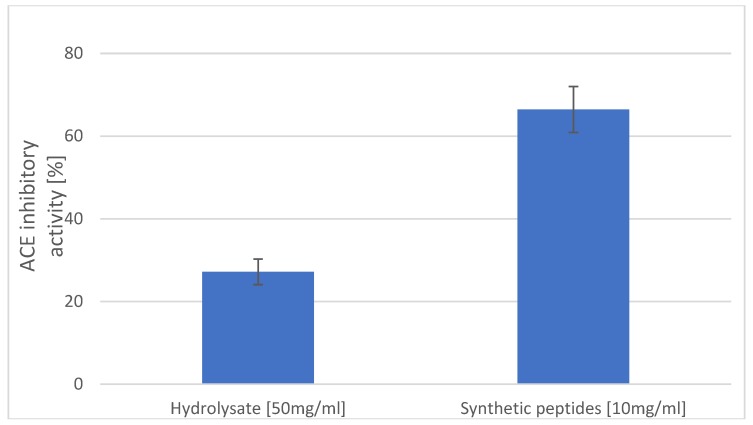
ACE-inhibitory activity [%] of hydrolysates and the synthetic Ala-Tyr peptide.

**Table 1 molecules-24-00097-t001:** The composition and amino acid profile of the hydrolysate of carp skin gelatin.

The Composition of Carp Skin Gelatin Hydrolysate	
Dry weigh [%]	90.00 ± 0.56
Protein [%]	80.09 ± 0.43
Fat [%]	0.93 ± 0.00
Ash [%]	4.48 ± 0.00
Amino acid profile (% of protein)	
Alanine (Ala)	11.33 ± 0.06
Arginine (Arg)	9.49 ± 0.05
Aspartic acid (Asp)	6.78 ± 0.04
Glutamic acid (Glu)	7.15 ± 0.04
Glycine (Gly)	28.12 ± 0.15
Histidine (His)	1.04 ± 0.01
Isoleucine (Ile)	1.41 ± 0.01
Leucine (Leu)	2.84 ± 0.02
Lysine (Lys)	4.16 ± 0.02
Methionine (Met)	2.30 ± 0.01
Phenylalanine (Phe)	2.19 ± 0.00
Proline + hydroxyproline (Prol + Hyp)	13.60 ± 0.07
Serine (Ser)	3.86 ± 0.02
Threonine (Thr)	3.05 ± 0.02
Tryptophan (Trp)	0.00 ± 0.00
Tyrosine (Tyr)	0.64 ± 0.00
Valine (Val)	2.59 ± 0.01

**Table 2 molecules-24-00097-t002:** DPPH radical—scavenging and metal-chelating activities of the fractions from gel filtration.

Fraction	DPPH Radical–Scavenging Activities [%]	Metal-Chelating Activity [%]
F1	n/d	5.36 ± 2.46
F2	6.03 ± 3.50	8.31 ± 0.86
F3	n/d	8.07 ± 4.32
F4	n/d	7.20 ± 1.56
F5	12.61 ± 2.27	9.05 ± 2.21
F6	12.93 ± 4.80	7.71 ± 4.06
F7	n/d	8.02 ± 4.01
F8	13.97 ± 7.04	5.90 ± 1.36

n/d not detected.

**Table 3 molecules-24-00097-t003:** Antioxidant properties of the hydrolysate and the synthetic Ala-Tyr peptide as well as their stability after in vitro digestion.

	Before In Vitro Digestion			After In Vitro Digestion		
	**FRAP [μM Trolox/mg Sample]**	**DPPH Scavenging** **[%]**	**Metal Chelating Activity** **[%]**	**FRAP [μM Trolox/mg Sample]**	**DPPH Scavenging** **[%]**	**Metal-Chelating Activity** **[%]**
Hydrolysate [50 mg/mL]	2.65 ^a^ ± 0.07	23.76 ^a^ ± 3.94	64.01 ^a^ ± 4.64	1.70 ^b^ ± 0.11	27.23 ^a^ ±2.07	64.31 ^a^ ± 9.70
Synthetic peptides [10 mg/mL]	89.32 ^a^ ± 1.80	38.69 ^a^ ± 3.09	3.09 ^b^ ± 0.15	62.25 ^b^ ± 1.07	34.47 ^a^ ± 4.33	73.25 ^a^ ± 7.08

^a.^, ^b.^: Results with different lowercase letters are significantly different (*p* < 0.05).

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
