# Peer review of "Identification of Antioxidant Peptides in Enzymatic Hydrolysates of Carp (Cyprinus Carpio) Skin Gelatin"

_molecules, 2018, doi:10.3390/molecules24010097_

Round 1

Reviewer 1 Report

Authors have fully addressed my previous concerns.

Author Response

We  would  like  to  thank  the  Reviewer  for   thoughtful  review  of  the  manuscript.  They  raise important issues and their inputs are very helpful for  improving the manuscript

Reviewer 2 Report

The authors faith into the action of a dipeptide is bordering on the absurd. More proof that the actual carp skin consumption is able to give actual ace inhibition is warranted.  

My comment is the choice of protein based ace inhibitor. The paper seems to
have too much faith in their assay. Where is the proof that a simple
dipeptide is enough to provide ace inhibition?

Author Response

Comment:

The authors faith into the action of a dipeptide is bordering on the absurd. More proof that the actual carp skin consumption is able to give actual ace inhibition is warranted.  My comment is the choice of protein based ace inhibitor. The paper seems to  have too much faith in their assay. Where is the proof that a simple  dipeptide is enough to provide ace inhibition?

Response:

We would like to thank the Reviewer for noting the insufficient amount of literature data confirming the ability to inhibit the ACE enzyme by the Alanine-Tyrosine dipeptide. Broad discussion on the subject was added in the text of the manuscript. We would also like to emphasize that activity inhibiting the ACE enzyme, dipeptides similar to those isolated in our research, is very well documented in literature (Schwab, A., Macerata, R., Rogers, W., Barton, J., Skiles, J., & Khandwala, A. (1984). Inhibition of angiotensin-converting enzyme by dipeptide analogs. Research Communications in Chemical Pathology and Pharmacology, 45(3), 339-345.

Wu, H., He, H. L., Chen, X. L., Sun, C. Y., Zhang, Y. Z., & Zhou, B. C. (2008). Purification and identification of novel angiotensin-I-converting enzyme inhibitory peptides from shark meat hydrolysate. Process Biochemistry, 43(4), 457-461.

 Nakahara, T., Sano, A., Yamaguchi, H., Sugimoto, K., Chikata, H., Kinoshita, E., & Uchida, R. (2009). Antihypertensive effect of peptide-enriched soy sauce-like seasoning and identification of its angiotensin I-converting enzyme inhibitory substances. Journal of Agricultural and Food Chemistry, 58(2), 821-827).

Our work even confirms other almost identical reports on the subject of such dipeptide activity. Food derived dipeptides are thought to diminish blood pressure mostly by inhibition of the angiotensin I converting enzyme (ACE)  in the cardiovascular system. However, the exact mechanisms underlying their vasoprotective actions, which may well extend beyond the established ACE inhibitory effects, have not been identified (Erdmann, Kati, et al. "The ACE inhibitory dipeptide Met-Tyr diminishes free radical formation in human endothelial cells via induction of heme oxygenase-1 and ferritin." The Journal of Nutrition 136.8 (2006): 2148-2152). Notwithstanding according to Reviewer suggested we decide delate the sentence about ACE in conclusion.

This manuscript is a resubmission of an earlier submission. The following is a list of the peer review reports and author responses from that submission.

Round 1

Reviewer 1 Report

This manuscript (molecules-401585) deals with antioxidant peptides in enzymatic hydrolysates of carp (Cyprinus carpio) skin gelatin.  Enzymatic hydrolysates was subjected to purification GFC and HPLC before the most active peptide was identified. ACE inhibitory activity and digestive stability was evaluated. This manuscript was interesting; however, some major concerns existed before drawing the conclusions. The originality of this manuscript was limited and methods used were not sound.

1. During hydrolysis of gelatin from carp skin, why 3h was chosen for enzymatic hydrolysis. Please give reasons.

2. Why FRAP of the peptide fractions was selected for screening antioxidant activity instead of the others?  What was the difference between methods? Please explain.

3. The method used for identification of the most active peptide should be added.

4. What was the purity of the synthetic peptide?

5. Line 252. I disagree the sentence ‘These results indicate that the hydrolysate and synthetic peptide Ala-Tyr effectively retain the overall antioxidant effect in the gastrointestinal tract.’ As authors only determined the antioxidant activity in vitro, the antioxidant activity in vivo remained to be determined.

6. Two important literature on antioxidant activity of gelatin hydrolysates were missing. Please comment them in discussion section.

Journal of Food Biochemistry (2017), 41(3), e12350.

Journal of Aquatic Food Product Technology,   (2015), 24(7), 648-660.

Reviewer 2 Report

the paper appears to describe a method to collect dipeptide from carp skin but the choice to characterize it's actions as having effects on ace inhibition, seems pretty far fetched. The conclusion would be more served by additional test with a similar fraction from some other fish.